# Developing functional markers for vitamin E biosynthesis in oil palm

Yajing Dou[1,2⊚], Wei Xia[1⊚], Annaliese S. Mason[3], Dongyi Huang[1], Xiwei Sun[2], Haikuo Fan[2], Yong Xiao[2,4]*

1 College of Tropical Crops, Hainan University, Haikou, Hainan, P.R. China, 2 Coconut Research Institute, Chinese Academy of Tropical Agricultural sciences, Wenchang, Hainan, P.R. China, 3 Plant Breeding Department, The University of Bonn, Bonn, North Rhine-Westphalia, Germany, 4 Sanya Research Institute, Chinese Academy of Tropical Agricultural Sciences, Sanya, Hainan, P.R. China

⊚ These authors contributed equally to this work.
* coconut_oilpalm@163.com, xiaoyong1980@catas.cn

**Data Availability Statement:** The raw datas of 24 transcriptomes (eight sample and three biological replicates) are available in the European Nucleotide Archive (accession number PRJEB38102, https://www.ebi.ac.uk/ena/browser/view/PRJEB38102).

## Abstract

Vitamin E is essential for human health and plays positive roles in anti-oxidation. Previously, we detected large variation in vitamin E content among 161 oil palm accessions. In this study, twenty oil palm accessions with distinct variation in vitamin E contents (171.30 to 1 258.50 ppm) were selected for genetic variation analysis and developing functional markers associated with vitamin E contents. Thirty-seven homologous genes in oil palm belonging to vitamin E biosynthesis pathway were identified via BLASTP analysis, the lengths of which ranged from 426 to 25 717 bp (average 7 089 bp). Multiplex PCR sequencing for the 37 genes found 1 703 SNPs and 85 indels among the 20 oil palm accessions, with 226 SNPs locating in the coding regions. Clustering analysis for these polymorphic loci showed that the 20 oil palm accessions could be divided into five groups. Among these groups, group I included eight oil palm accessions whose vitamin E content (mean value: 893.50 ppm) was far higher than other groups (mean value 256.29 to 532.94 ppm). Correlation analysis between the markers and vitamin E traits showed that 134 SNP and 7 indel markers were significantly ($p < 0.05$) related with total vitamin E content. Among these functional markers, the indel *EgTMT-1-24* was highly correlated with variation in vitamin E content, especially tocotrienol content. Our study identified a number of candidate function associated markers and provided clues for further research into molecular breeding for high vitamin E content oil palm.

## Introduction

Oil palm (*Elaeis guineensis*, 2n = 32) is the most efficient oil crop in the world, which has the highest oil yield per unit area of all oil crops [1, 2]. Oil palm occupies about eight percent of total plantation area of oil crops worldwide [3], and the palm oil contributes more than 30% of total vegetable oil production in the world [4, 5]. Palm oil obtained from the oil palm mesocarp is rich in vitamins, especially for vitamin A and E [6, 7]. In nature, vitamin E is comprised of four tocopherol types ($\alpha$, $\beta$, $\gamma$, and $\delta$) and four tocotrienol types ($\alpha$, $\beta$, $\gamma$, and $\delta$) [8–10].

Also, the raw datas of multiple PCR sequencing of 20 oil palm individuals have aslo been uploaded to ENA (accession number PRJEB38216, https://www.ebi.ac.uk/ena/browser/view/PRJEB38216).

**Funding:** This work was supported by Central public-interest scientific institution basal research fund for the Chinese Academy of Tropical Agricultural Sciences (No. 1630152017008 and 1630152019002) and the fund for species and varieties conservation of sector project of the Ministry of Agriculture and Rural Affairs (1820039).

**Competing interests:** The authors have declared that no competing interests exist.

Tocopherols are mainly found in the leaves and seeds of dicotyledonous plants [11, 12], whereas tocotrienols are found in the seed and endosperm tissues of monocotyledonous plants, especially in cereal grain crops such as rice, maize, and barley [8, 13, 14]. Previous research had showed that palm oil had high vitamin E content (approximately 600–1 000 ppm), which is comprised of $\alpha$-tocopherol, $\alpha$-tocotrienol, $\gamma$-tocotrienol, and $\delta$-tocotrienol [15, 16]. Of these components, $\alpha$-tocopherol occupies a small percentage of total vitamin E content (10%–30%), while the tocotrienols are the major components (70%–90%) [7]. In fact, palm oil has the highest content of tocotrienols in all major types of edible oil [17, 18]. Vitamin E is a powerful anti-oxidant, tocotrienols in which are proven to have 40–60 times more potency than $\alpha$-tocopherol [19, 20]. Due to this property, palm oil has excellent stability at high temperature and the food prepared with palm oil can have long shelf life [21]. Besides anti-oxidant [22], tocotrienols also have the functions of neuroprotection [23], anti-cancer [24], cardio-protective [25], anti-inflammatory [26], anti-diabetic [6], and inhibition of cholesterol synthesis [27], which make palm oil beneficial for human health. Vitamin E in oil palm has been extensively studied for its nutritional and health properties, attributed largely to its high tocotrienols content [28]. Our previous study has detected large phenotypic variation for vitamin E content in 161 oil palm accessions and the average value of total vitamin E content varied from 172.45 to 1 287.96 ppm [16], which could provide genetic resources for identifying elite alleles associated with high vitamin E content [16]. However, as a perennial tree, oil palm requires 10–19 years for phenotypic selection by traditional breeding method [29]. Molecular breeding uses traits linked markers for seedling pre-selection of desired vitamin E phenotypes, which has a potential to greatly shorten the breeding cycle and reduce costs.

Single nucleotide variations in coding sequence (CDS) regions can result in missense and nonsense mutation, causing influencing on gene function and variations in agronomic traits [30–33]. For example, four non-synonymous substitutions for soybean Terminal Flower 1 (*GmTfl1*) lead to changes in stem growth habit and improvement of soybean yield [34, 35]. A single nucleotide polymorphism SNP in a C2H2-type transcription factor increased broad-spectrum blast resistance in rice [36]. Two nucleotide mutations in a serine hydroxymethyl-transferase (SHMT) gene improved soybean resistance to cyst nematode [37]. Polymorphic sites within genes that are related to trait variation could be used to develop functional markers, which are beneficial to marker-assisted selection [38, 39].

Approaches to develop functional markers in plant species include polymerase chain reaction (PCR) based markers, such as simple sequence repeats (SSRs) [40], sequence tagged site (STS) [41], expressed sequence tag-derived microsatellite (EST-SSR) [42, 43], and intron sequence amplified polymorphism (ISAP) [44, 45]. Recently, a single-PCR-based approach was used to develop single nucleotide polymorphism (SNP) markers by performing PCR amplification for specific gene locus from different accessions [46–48]. However, this approach is inefficient and high cost, for obtaining genotypic information for one locus at a time. Multiplex PCR sequencing offers a better solution to produce markers from multiple loci, in which, pairs of primers covered different genic regions are used in amplification and PCR products are sequenced using next-generation sequencing to identify SNP markers [49]. This technology has been validated as a powerful tool to develop molecular markers for targeted sequences.

Functional markers development relies on the knowledge of genes with assigned functions [38]. Extensive research had revealed the vitamin E biosynthesis pathways (Fig 1), where vitamin E is mainly synthesized via shikimate pathway and methylerythritol phosphate (MEP) pathway [10]. A phytol recycling pathway for phytyl-PP production from chlorophyll degradation also partakes [9]. The genes encoding the respective enzymes of these pathways have been identified and cloned in *Arabidopsis thaliana* [8, 50–52] and some other plant species [53, 54]. The functions of most genes in biosynthesis pathway also have been verified by transgenic

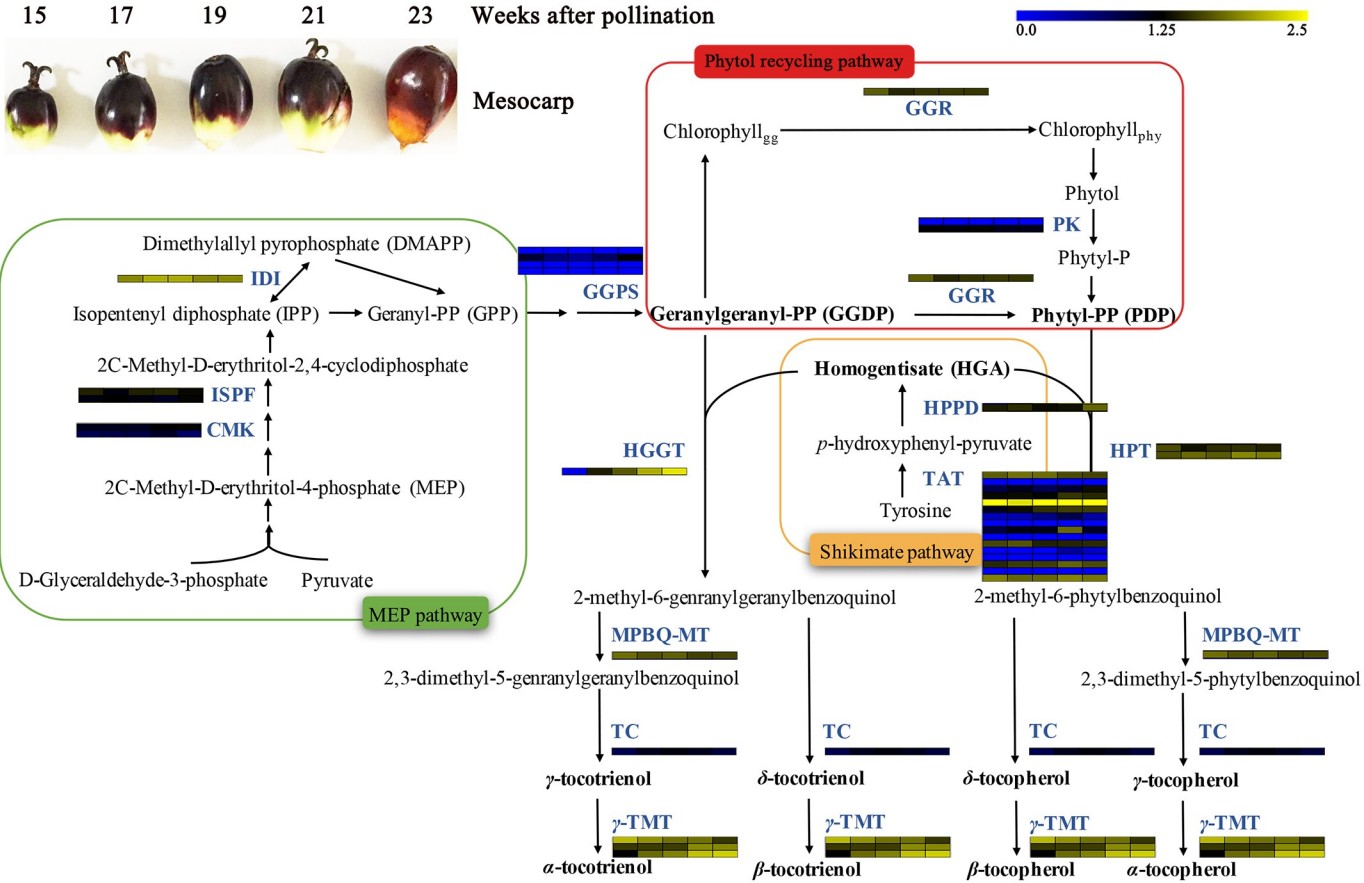

**Fig 1. Biosynthesis pathway of vitamin E and the expression pattern of candidate genes involved in the pathway.** The heatmaps of candidate genes were visualized by the MeV software. The expression levels were calculated by using $Log_2$ (FPKM+1). Abbreviations: CMK, 4-diphosphocytidyl-2-C-methyl-D-erythritol kinase; HGGT, Homogentisate geranylgeranyltransferase; HPPD, 4-hydroxyphenylpyruvate dixygenase; HPT, homogentisate phytyltransferase; IDI, isopentenyl diphosphate isomerase; ISPF, 2-C-methyl-D-erythritol-2,4-cyclodiphosphate synthase; GGPS, geranylgeranyl pyrophosphate synthase; GGR, geranylgeranyl reductase; MPBQ-MT, 2-methyl-6-phytyl-1,4-hydroquinone methyltransferase; PK, phytol kinase; TAT, tyrosine aminotransferase; TC, tocopherol/tocotrienol cyclase; TMT, tocopherol/tocotrienol methyltransferase.

technology, such as *HPPD* [55, 56], *HPT* [57, 58], *HGGT* [59, 60], *MPBQ-MT* [61], *TC* [62] and *γ-TMT* [63], which are conducive to finding candidate genes belonging to vitamin E biosynthesis pathways in oil palm.

In recent years, genetic variations associated with total vitamin E or component contents have been identified, which could be used for markers development and molecular breeding. Fritsche *et al*. [64] identified two SNPs in *MPBQ-MT* (*VTE3*) gene associated with γ-tocopherol content and α-tocopherol content separately, as well as one SNP in *HPT* (*VTE2*) gene associated with γ-tocopherol content in rapeseed. Two markers—InDel7 and InDel118—derived from insertion/deletions (Indels) polymorphism located respectively within the 5' UTR region and the promoter region of *Zea mays γ-TMT* (*VTE4*) gene were significantly associated with α-tocopherol levels in maize kernels [65]. The effects of these two causative indels in *ZmVTE4* gene were validated by haplotype analysis [66]. However, few researches were reported on the genetic dissection of vitamin E biosynthesis in oil palm. Kong *et al*. [13] successfully cloned *HGGT* and *HPT* genes from two oil palm species and found that these two genes had a high expression level in oil palm mesocarp. Meanwhile, *EgHGGT* gene was validated to be associated with the biosynthesis of α-tocotrienol and total vitamin E content [16].

Based on the large phenotypic variations of vitamin E content and the known candidate genes involved in vitamin E biosynthesis in oil palm, it is possible to develop functional markers associated with vitamin E content. In this study, 20 oil palm accessions with distinct variation of vitamin E content (171.30 to 1 258.50 ppm) were selected from the 161 accessions employed by Luo *et al*. [16] to develop trait related functional markers. Subsequently, we analyzed the correlation between these functional markers and vitamin E content, providing functional markers for further molecular breeding of oil palm cultivars with high vitamin E content.

## Materials and methods

### Plant materials, vitamin E investigation and DNA extraction

In previous studies, the vitamin E content of 161 oil palm accessions with three biological and technical replicates per sample were measured by high-performance liquid chromatography (HPLC) analysis [16]. In this study, ten top oil palm accessions with high total vitamin E content (more than 650 ppm) and ten top accessions with low total vitamin E content (less than 500 ppm) were selected from the 161 oil palm accessions. The 20 oil palm accessions were planted in 2008 in the oil palm germplasm collection of the Coconut Research Institute of Chinese Academy of Tropical Agricultural Sciences, Wenchang, Hainan, China. The average temperature and humidity were approximately 23.9˚C and 89%, respectively. Detailed information for these 20 oil palm accessions is listed in S1 Table.

The spear leaves of the 20 oil palm accessions were sampled for DNA extraction using a modified cetyltrimethyl ammonium bromide (CTAB) protocol [67]. The procedure was as follows: about 0.2 g of leaf tissue was ground into a fine powder with liquid nitrogen using a mortar and pestle. The pulverized tissue was immediately transferred to a 2 mL tube with 1 mL extraction buffer (119.8 g/L sucrose; 0.1 mol/L Tris-HCl; 5 mmol/L EDTA; 20 g/L PVP 40) and 30 µL $\beta$-mercaptoethanol. This mixture was then vortexed vigorously and centrifuged for 5 minutes at 8 000 rpm at 4˚C, and the supernatant was discarded. Further, 1 mL of 65˚C lysis buffer (0.1 mol/L Tris-HCl; 81.8 g/L NaCl; 20 mmol/L EDTA; 20 g/L CTAB; 20 g/L PVP 40) was added in each tube, the sample was vortexed thoroughly, incubated in 65˚C water bath for 60 minutes and shook once every ten minutes, then centrifuged for 10 minutes at 8 000 rpm at room temperature. The supernatant was transferred to a new 2 mL tube, mixed with an equal volume of chloroform and isoamyl alcohol solution (24:1, v/v), vortexed gently, centrifuged at 12 000 rpm for 18 minutes, and the aqueous phase (top layer) was transferred into a new tube. This step was repeated twice. After that, 1/3 volume of 3 mol/L sodium acetate (pH = 5.2) and double volume of chilled isopropanol were added, mixed by inverting the tube 20–30 times. DNA precipitation could be seen in the form of a cottony white colored mass in the solution, then incubated at -20˚C overnight. The next day, the DNA was pelleted by centrifugation at 8 000 rpm for 3 minutes and washed twice with 1 mL of 70% ethanol. After removing all residual ethanol, the pellet was air-dried for 20–30 minutes and the obtained DNA was dissolved with 300 µL TE buffer. The concentration and quality of the 20 oil palm DNA samples was examined using a Nanodrop 2000 spectrophotometer (NanoDrop Technologies, Wilmington, DE, USA). The quantified DNA was diluted to 100 ng/µL for PCR amplification.

### Identification of candidate genes involved in vitamin E biosynthesis

All protein-coding gene sequences of *E. guineensis*, *A. thaliana*, and *Z. mays* were downloaded from the National Center for Biotechnology Information website (NCBI). Putative genes belonging to the vitamin E biosynthesis pathway from *A. thaliana* and *Z. mays* were aligned against the oil palm protein database with a cut-off e-value of 1e-10 [68–71] to identify the

best-hit homologous genes in oil palm. The conserved domains of these homologous genes were predicted by aligning with the pfam database (http://pfam.xfam.org/search). Gene structures were predicted by aligning protein sequences with the *E. guineensis* genome and drawn using the software GSDS 2.0 (http://gsds.cbi.pku.edu.cn).

## FPKM calculations for transcriptomes obtained from different developmental stages of oil palm mesocarps

A total of 5 transcriptomes (single-end, 454 GS FLX Titanium), including 15-week-old mesocarp (SRR190698), 17-week-old mesocarp (SRR190699), 19-week-old mesocarp (SRR190700), 21-week-old mesocarp (SRR190701), and 23-week-old mesocarp (SRR190702) [2], were downloaded from the NCBI website. FPKM (fragments per kilobase of exon model per million mapped reads) was calculated to measure gene expression level by using the following formula [72, 73]:

$$FPKM = \frac{10^6 C}{NL/10^3}$$

where C is the number of fragments which exclusively aligned to one expressed sequence; N is the total number of reads which are aligned to all expressed sequences and L is the basic number in the CDS of the corresponding expressed sequence.

## Primer design and PCR amplification

A total of 328 primer pairs were designed to cover the complete genomic region of the candidate genes in the vitamin E biosynthesis pathway by using the software Primer Premier 5.0. These primer pairs were used to amplify intronic and exonic regions with amplicons ranging in length from 377 to 1 709 bp. The primer sequences are listed in S2 Table. We used a 20 μL PCR reaction system that contained 3 μL DNA (50 ng/μL), 2 μL pfu PCR Buffer (10 ×), 0.4 μL dNTP (10mM), 0.4 μL forward primer (10 mM), 0.4 μL reverse primer (10 mM), 0.4 μL pfu DNA polymerase, and 13.4 μL ddH$_2$O. The PCR program was set as follows: 94°C for 30 s, 30 cycles of 94°C for 10 s, 50–60°C for 30 s, and 72°C for 60 s, and then 72°C for 10 minutes. The PCR products were electrophoretically visualized on a 1% agarose gel.

## DNA library construction and illumina sequencing

PCR products per sample were mixed and fragmented with a Bioruptor Pico Sonication Device (Diagenode, Liege, Belgium). The fragmented DNA was detected by electrophoresis and targeted sizes (500 bp) were selected for DNA library construction which was conducted with TrueLib DNA Library Rapid Prep Kit (Excell Biotech, Taicang, Jiangsu, China, catalogue number NGS00-1063). End Rapid Repair Reaction Buffer (EB buffer) was used to resolve these short fragments for end reparation and poly (A) addition in a 50 μL reaction system that contained 1 μg fragmented DNA, 10 μL 5 × EB buffer, 3 μL end prep enzyme mix and ddH$_2$O. These treated DNA fragments were used for the following PCR amplification using a two-step PCR program: 22°C for 10 minutes and 72°C for 10 minutes. The sequence adaptors were linked to two ends of short DNA sequences and the adaptor-ligated DNA fragments were purified using AMPure XP beads (Beckman Coulter Inc, Brea, CA, USA, catalogue number A63881). The purified products were selected as templates for PCR amplification using ExHiFi 2× PCR Master Mix, Universal primer and Index primer in a 50 μL reaction system. After PCR products were purified (AMPure XP beads), the concentration and quality of DNA library were assessed with Qubit 2.0 Flurometer (Life Technologies, Carlsbad, CA, USA) and

Agilent Bioanalyzer 2100 system separately. Finally, the prepared DNA library was sequenced on an Illumina Hiseq 4000 platform (Illumina Inc., San Diego, CA, USA) and paired-end reads were generated by sequencing. The sequence quality of raw reads was checked using the FastQC software (http://www.bioinformatics.babraham.ac.uk/projects/ fastqc/). Raw reads were filtered with the Trimmomatic software [74] to obtain clean reads.

## SNP and indel genotyping

The genome sequence of *E. guineensis* was downloaded from the NCBI website (RefSeq: GCF_000442705.1). Clean reads were mapped onto the reference genome and selected to assemble gene sequences by using the software Vevet [75]. The scaffolds mapped to the candidate gene regions were further assembled via CAP3 [72] and manually checked in MEGA 7 [76]. Coverage ratios of assembled sequences mapped on reference sequences were calculated. SNPs and indels were manually identified after multiple sequence alignments for candidate gene sequences from the 20 oil palm accessions via MEGA 7. The exons sequences of the assembled genes were extracted and translated into amino acid sequences, and then the amino acid variations were manually counted.

The Pearson's correlation coefficient and significance between markers and vitamin E content was performed by using the software IBM SPSS Statistics Version 20–32 bit.

## Clustering analysis

The representative sequence for each gene with SNP and indel variations from each oil palm accession was combined into a single sequence for cluster analysis. Phylogenitic analysis was performed according to the procedures described by Bast [77] using the software MEGA 7, where the combined sequences were subjected to multiple sequence alignments by ClustalW [78]. Phylogenetic tree was generated by maximum likelihood method based on general time reversible (GTR) model [79] with 500 bootstrap repetitions. Pairwise distances between sequences were inferred under the maximum composite likelihood (MCL) approach. Meanwhile, the heatmap of vitamin E content was drawn using the software MeV 4.8 [80].

## RNA extraction, transcriptome sequencing and expression analysis

Eight oil palm individuals (three biological replicates) with distinct variation of vitamin E contents were subjected to transcriptome sequencing, including R17 (ERR4091630, ERR4091631, ERR4091632), R15 (ERR4091624, ERR4091625, ERR4091626), R12 (ERR4091618, ERR4091619, ERR4091620), R06 (ERR4091609, ERR4091610, ERR4091611), R04 (ERR4091606, ERR4091607, ERR4091608), R07 (ERR4091612, ERR4091613, ERR4091614), R09 (ERR4091615, ERR4091616, ERR4091617), and R14 (ERR4091621, ERR4091622, ERR4091623). Mature mesocarps were sampled from these eight oil palm accessions (three replicates per sample) and immediately frozen in liquid nitrogen. Total RNA was extracted separately using the MRIP method described by Xiao *et al.* (2012) [81]. RNA degradation and contamination, especially for DNA contamination, was detected by 1.5% agarose gels. RNA concentration and purity were measured using the NanoDrop 2000 Spectrophotometer. RNA integrity was assessed by using the RNA Nano 6000 Assay Kit of the Agilent Bioanalyzer 2100 (Agilent Technologies, Palo Alto, CA, USA). mRNA was enriched by Oligo(dT) magnetic beans and then rRNA was removed. The purified mRNA was fragmented with divalent cations under increased temperature. These short fragments were taken as templates to synthesize the first-strand cDNA using random hexamer primers and superscript™ III (Invitrogen™, Carlsbad, CA, USA). Second-strand cDNA was then synthesized in a solution containing buffer, dNTP, RNaseH and DNA polymerase I, and subsequently purified using a QiaQuick PCR extraction

kit (Qiangen). EB buffer was used to resolve these short fragments for end reparation and poly (A) addition. The sequence adaptors were linked to two ends of short cDNA sequences and suitably sized (100–200 bp) cDNA fragments were selected out for PCR amplification based on the agarose gel electrophoresis results. Finally, the library established was sequenced with an Illumina Hiseq™ 2000 (Illumina Inc., San Diego, CA, USA). The paired-end library was developed according to the protocol of the Paired-End Sample Preparation Kit (Illumina, USA). FPKM values were calculated to investigate the expression levels of the *EgTMT-1* gene in different oil palm individuals. Pearson's correlation coefficient between FPKM values and vitamin E traits were calculated using the software IBM SPSS Statistics.

## Results

### Classification of 20 oil palm accessions based on total vitamin E content

According to the vitamin E contents determined by Luo *et al.* [16], the selected 20 oil palm accessions were classified into four groups based on their total vitamin E content (Fig 2). Six oil palm accessions (H group) showed high vitamin E content, ranging from 1 031.40 to 1 258.50 ppm with an average value of 1 122.67 ppm, including R10, 08, 07, 11, 09 and 14. Four oil palm accessions (M group) showed medium vitamin E content, ranging from 656.46 to 929.35 ppm with an average value of 771.66 ppm, including R03, 04, 02, and 19. Moreover, five oil palm accessions (L group) showed low vitamin E content, ranging from 308.19 to 422.62 ppm with an average value of 373.87 ppm. The remaining five oil palm accessions (EL group) showed extremely low vitamin E content, ranging from 171.30 to 282.58 ppm with an average value of 244.05 ppm, including R20, 17, 13, 05, and 16. The average vitamin E content of these 20 oil palm accessions was 645.61 ± 376.41 ppm (mean ± SD).

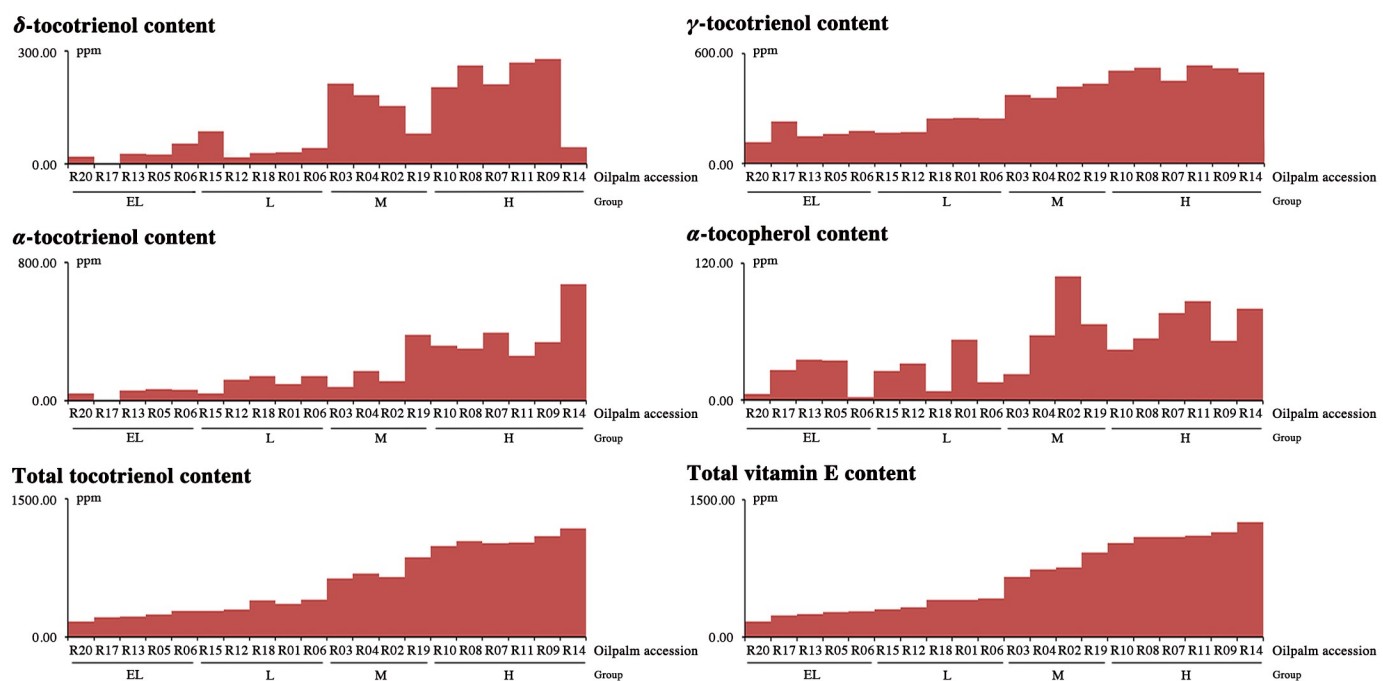

**Fig 2. Variations of vitamin E content in 20 oil palm accessions.** The vitamin E contents in 20 oil palm accessions (three biological and technical replicates per sample) included δ-tocotrienol, γ-tocotrienol, α-tocotrienol, α-tocopherol, total tocotrienol, and total vitamin E.

## Identification of candidate genes related to vitamin E biosynthesis

Thirty-seven candidate genes involved in vitamin E biosynthesis were identified by BLAST analysis, where the best-hit genes were found by blasting the protein database of *E. guineensis* against the *Arabidopsis* and *Z. mays* genes belonging to vitamin E biosynthesis pathway (Table 1). A total of 17 *EgTAT* genes were identified that catalyze tyrosine into *p*-hydroxyphenyl-pyruvate. Compared to the three *TAT* genes present in *Arabidopsis*, a significant expansion of *EgTAT* copies (up to 17) was detected in the genome of *E. guineensis*. One *EgHPPD* gene was identified, which converses *p*-hydroxyphenyl-pyruvate to HGA. Meanwhile, one *EgHGGT*

**Table 1. Predicted candidate genes involved in vitamin E biosynthesis in *E. guineensis*.**

| Gene name | Best-hit gene in *A. thaliana* | E-value | Chromosome | Strand | Gene ID | Gene length (bp) | Protein ID | Protein length (aa) |
|---|---|---|---|---|---|---|---|---|
| *EgCMK-1* | *AtCMK* | 0 | chr5 | + | LOC105045599 | 7 589 | XP_010922252.1 | 397 |
| *EgCMK-2* | *AtCMK* | 2E-151 | chr10 | - | LOC105052446 | 5 083 | XP_029122665.1 | 397 |
| *EgHPPD* | *AtHPPD* | 2E-80 | Scaffold | - | LOC105035707 | 5 743 | XP_019702972.1 | 200 |
| *EgHPT-1* | *AtHST* | 2E-180 | chr6 | - | LOC105046456 | 7 259 | XP_010923356.1 | 402 |
| *EgHPT-2* | *AtHPT1* | 4E-173 | chr8 | + | LOC105050383 | 18 852 | NP_001291355.1 | 400 |
| *EgHGGT* | *AtHPT1* | 2E-125 | chr10 | - | LOC105052944 | 7 191 | XP_010932231.1 | 462 |
| *EgIDI* | *AtIDI2* | 7E-46 | Scaffold | + | LOC105061124 | 940 | XP_010943386.1 | 104 |
| *EgGGR* | *AtGGR* | 9E-107 | Scaffold | - | LOC105032472 | 1 592 | XP_010905229.1 | 413 |
| *EgGGPS-1* | *AtGGPS2* | 6E-28 | Scaffold | + | LOC105037082 | 426 | XP_010911086.1 | 141 |
| *EgGGPS-2* | *AtGGPS1* | 2E-38 | chr1 | - | LOC105048867 | 11 386 | XP_010926643.1 | 395 |
| *EgGGPS-3* | *AtGGPS1* | 3E-130 | chr3 | + | LOC105041045 | 3 041 | XP_010916141.1 | 349 |
| *EgGGPS-4* | *AtGGPS1* | 6E-144 | chr15 | + | LOC105057980 | 1 755 | XP_010939037.1 | 369 |
| *EgISPF-1* | *AtISPF* | 3E-102 | chr4 | + | LOC105043318 | 6 681 | XP_010919124.2 | 236 |
| *EgISPF-2* | *AtISPF* | 9E-105 | chr11 | + | LOC105054207 | 3 774 | XP_010933967.1 | 208 |
| *EgMPBQ-MT* | *AtMPBQ-MT* | 0 | chr1 | + | LOC105055609 | 6 402 | XP_010935791.1 | 340 |
| *EgPK-1* | *AtPK* | 3E-112 | Scaffold | + | LOC105033444 | 4 138 | XP_010906565.1 | 323 |
| *EgPK-2* | *AtPK* | 3E-70 | Scaffold | - | LOC105036025 | 6 539 | XP_010910062.1 | 307 |
| *EgTAT-2* | *AtTAT2* | 5E-22 | Scaffold | - | LOC105033000 | 2 850 | XP_010905916.1 | 474 |
| *EgTAT-3* | *AtTAT2* | 2E-16 | chr1 | + | LOC105050532 | 9 153 | XP_010928898.1 | 481 |
| *EgTAT-4* | *AtTAT2* | 2E-24 | chr2 | - | LOC105037948 | 4 940 | XP_010911890.1 | 412 |
| *EgTAT-5* | *AtTAT2* | 0 | chr3 | + | LOC105040940 | 5 094 | XP_010916004.1 | 425 |
| *EgTAT-6* | *AtTAT2* | 2E-14 | chr3 | - | LOC105040248 | 10 327 | XP_010914992.1 | 396 |
| *EgTAT-7* | *AtTAT1* | 2E-18 | chr3 | + | LOC105041987 | 8 295 | XP_019704422.1 | 477 |
| *EgTAT-8* | *AtTAT2* | 5E-15 | chr4 | + | LOC105043536 | 3 010 | XP_010919415.1 | 492 |
| *EgTAT-9* | *AtTAT3* | 6E-18 | chr7 | - | LOC105047871 | 15 209 | XP_010925286.1 | 396 |
| *EgTAT-10* | *AtTAT2* | 5E-17 | chr7 | - | LOC105048002 | 2 155 | XP_010925479.1 | 441 |
| *EgTAT-11* | *AtTAT3* | 8E-16 | chr8 | + | LOC105049882 | 10 084 | XP_010927963.1 | 481 |
| *EgTAT-12* | *AtTAT2* | 1E-16 | chr11 | - | LOC105054176 | 5 290 | XP_010933924.1 | 530 |
| *EgTAT-13* | *AtTAT2* | 3E-14 | chr13 | + | LOC105055886 | 2 139 | XP_010936199.1 | 460 |
| *EgTAT-14* | *AtTAT2* | 1E-18 | chr13 | + | LOC105056662 | 7 017 | XP_010937250.1 | 526 |
| *EgTAT-15* | *AtTAT3* | 4E-17 | chr15 | - | LOC105058291 | 2 319 | XP_010939473.1 | 468 |
| *EgTAT-16* | *AtTAT2* | 5E-28 | chr15 | - | LOC105058225 | 11 240 | XP_010939394.1 | 464 |
| *EgTAT-17* | *AtTAT3* | 4E-17 | chr15 | + | LOC105058582 | 7 350 | XP_010939849.1 | 475 |
| *EgTC* | *AtTC* | 0 | chr3 | - | LOC105040851 | 19 129 | XP_010915874.1 | 492 |
| *EgTMT-1* | *AtTMT* | 3E-153 | Scaffold | - | LOC105033221 | 25 717 | XP_010906230.1 | 327 |
| *EgTMT-2* | *AtTMT* | 2E-12 | chr4 | + | LOC105042568 | 1 649 | XP_010918151.1 | 369 |
| *EgTMT-3* | *AtTMT* | 1E-15 | chr6 | - | LOC105046511 | 11 273 | XP_010923409.1 | 342 |

and two *EgHPT* were identified. The protein sequences of EgHGGT and EgHPT-2 had higher similarity (56%), although the two types of genes have different functions in vitamin E biosynthesis. Moreover, one *MPBQ-MT* was identified, which converts 2-methyl-6-genranylgeranyl-benzoquinol (MGGBQ) and 2-methyl-6-phytyllbenzoquinol (MPBQ) to 2,3-dimethyl-5-genranylgeranylbenzoquinol and 2,3-dimethyl-5-phytybenzoquinol, respectively. Only one *EgTC* gene (the final step in the biosynthesis of $\gamma$-tocotrienol, $\delta$-tocotrienol, $\gamma$-tocopherol, and $\delta$-tocopherol) was identified. In addition, three *EgTMT* genes (involved in the production of $\alpha$-tocotrienol, $\beta$-tocotrienol, $\alpha$-tocopherol, and $\beta$-tocopherol) were identified.

The lengths of candidate genes involved in vitamin E biosynthesis varied from 426 to 25 717 bp, with an average length of 7 089 bp per candidate gene. Their protein sequences varied from 104 to 530 amino acids (aa), with an average length of 388 amino acids per gene.

## Candidate gene structures and conserved domains

Gene structures of the 37 candidate genes were visualized by the software GSDS (Fig 3). The exon numbers of these candidate genes varied from 1 to 15, among which, *EgTAT* contained the highest amount of exons (15), while *EgTMT-2* contained only one exon. *EgTAT* had 17 gene copies, some of which shared the same amount of introns and exons. For example, five *EgTAT* copies (*EgTAT-8*, *10*, *12*, *13*, *15*) contained three introns and four exons. Meanwhile, two *EgCMK*, two *EgISPF*, and two *EgPK* also had the same amount of introns and exons.

Conserved gene structures indicated that noticeable gene structure variations were existence in different *EgTMT* copies: *EgTMT-1* only contained one exon; *EgTMT-2* had 10 exons, and *EgTMT-3* had 13 exons. This gene structure variation may result in the divergence of *EgTMT* gene function. Some *EgTMTs* may be responsible for the transformation from $\gamma$-tocopherol and $\delta$-tocopherol to $\alpha$-tocopherol and $\beta$-tocopherol, while others may be responsible for the transformation from $\gamma$-tocotrienol and $\delta$-tocotrienol to $\alpha$-tocotrienol and $\beta$-tocotrienol. Meanwhile, *EgHGGT* and two *EgHPT* copies have similar protein sequence and showed different function, which also showed high variation in gene structure.

The functional domains analysis demonstrated that most candidate genes contained the corresponding conserved functional domain, except for *EgPK-1* and *EgPK-2*. Detailed information on candidate gene functional domains is listed in S3 Table.

## Expression pattern of candidate genes related to vitamin E biosynthesis in different developmental stages of mesocarp

In order to detect the expression pattern of candidate genes in different stages of oil palm mesocarp, we calculated the FPKM values based on transcriptoms derived from different developmental stages of mesocarps, including mesocarp 15 weeks (SRR190698), 17 weeks (SRR190699), 19 weeks (SRR190700), 21 weeks (SRR190701), and 23 weeks (SRR190702). Sixteen *EgTAT* genes showed different expression levels in the oil palm mesocarp. Among them, *EgTAT5* have high expression level in different stages of oil palm mesocarp, which also has high similarity (E-value = 0) with *Arabidopsis TAT* gene. *EgHPPD* and *EgHGGT* have higher expression level in mesocarp 21 weeks and 23 weeks than in other earlier stages of mesocarp. The two *EgHPT* showed different expression summits amomg mesocarp development: *EgHPT-1* has highest expression level in mesocarp 15 weeks, whereas *EgHPT-2* has highest expression level in mesocarp 21 weeks and 23 weeks. Meanwhile, Three *EgTMT* genes have all high expression level in any five mesocarp developmental stages. Moreover, the expression level of *EgTMT-1* was gradually decreased from mesocarp 15 weeks to 23 weeks, and *EgTMT-2* and *EgTMT-3* was gradually increased from mesocarp 15 weeks to 23 weeks. The expression patterns of all candidate genes involved in vitamin E biosynthesis were showed in Fig 1.

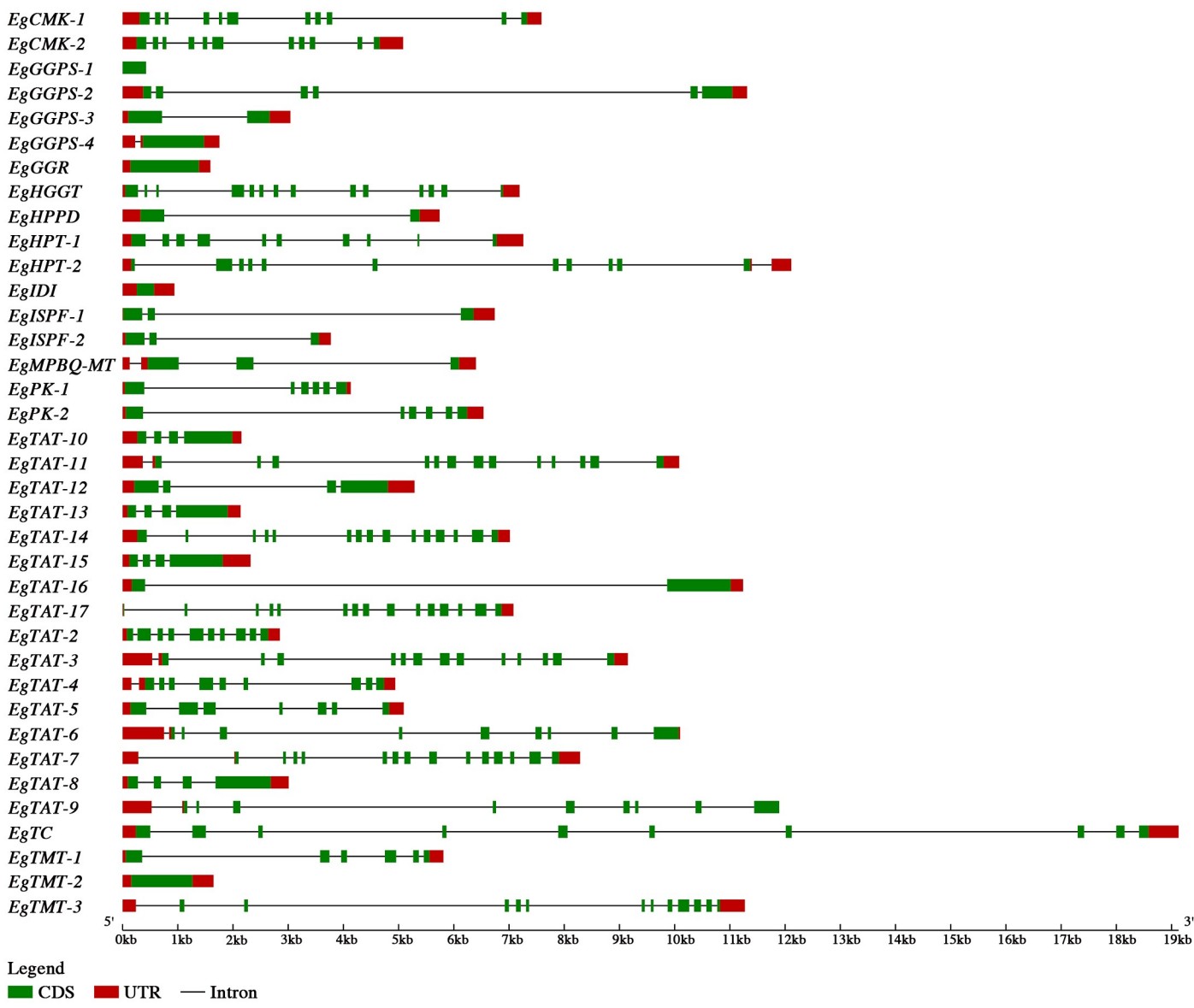

**Fig 3. Gene structure of 37 candidate genes involved in vitamin E biosynthesis in *E. guineensis*.**

## Developing functional molecular markers from the 20 oil palm accessions

PCR products per oil palm accession were produced and sequenced using the Illumina sequencing platform at an average sequencing depth of 906×. In total, the number of clean reads generated from the 20 DNA libraries ranged from 3 305 260 to 5 346 160, with an average of 4 268 221 ± 472 954 (mean ± SD) of per sample, and GC contents changed from 38.69% to 42.85%, Q30 values of clean reads were all above 95%. After assembly, 485 051 to 1 034 321 bp scaffolds were mapped to the oil palm reference genome. Subsequently, the scaffolds were mapped to the corresponding target genes involved in vitamin E biosynthesis and further assembled. Five genes (*EgGGR*, *EgGGPS-1*, *EgGGPS-3*, *EgGGPS-4*, *EgTMT-2*) were not sequenced well in all 20 oil palm accessions and was not further investigated in the present study. The amplified regions of the remaining 32 genes covered between 59.21% and 100% of

the corresponding targeted regions and the assembled fragments covered between 21.99% and 100% of amplified regions with an average of 14.02% missing (S4 Table).

The sequence alignment and analysis found 1 788 polymorphic sites (1 703 SNPs and 85 indels) among the 20 oil palm accessions. The density of SNP and Indel (the average length of fragments divided by the number of variations) of every gene was calculated and an average density of 1 SNP/179 bp and 1 indel/2 273 bp were observed (S4 Table). Of these, a large proportion of SNPs (1 410) and indels (74) were distributed in the intron regions of these candidate genes, while 67 SNPs and 11 indels were located in the untranslated regions and 226 SNPs in the coding sequence (CDS) regions (S5 and S6 Tables). Of the 226 SNPs within CDS regions, a total of 96 single nucleotide variations resulted in amino acid changes, which were mapped into 25 candidate genes. Meanwhile, among 96 SNPs, 56 were mapped into conserved domains of 16 candidate genes, including *EgHGGT*, *EgTAT1*, *EgTAT2*, *EgTAT4*, *EgTAT5*, EgTAT6, *EgTAT7*, *EgTAT8*, *EgTAT9*, *EgTAT10*, *EgTAT12*, *EgTAT14*, *EgTAT15*, *EgTAT16*, *EgMPBQ-MT*, *EgTC* (S1 Fig). No polymorphic sites caused frame-shift mutations in these genes among the 20 oil palm accessions. Among all candidate genes, *EgTMT-1* was the most polymorphic (295 SNPs and 20 indels), followed by the *EgTC* gene which harbored 142 SNPs and 7 indels.

## The association between genetic variants in vitamin E synthesis genes and vitamin E content

A total of 1 788 variations from 32 candidate genes in each oil palm accessions were combined into a sequence, and the length of the combined sequence was 3 391 bp. The 20 oil palm accessions were divided into five clusters (groups I to V) on the bases of the pairwise sequence similarity over 70% [82] and bootstrap values higher than 50% [83]. Group I included the most oil palm accessions (8) and had the highest vitamin E content (274.05 ppm to 1 147.77 ppm, average 893.50 ppm), which also included the above high vitamin content group (5, H group) and middle vitamin E content group (2, M group) except R05 (Fig 4). The four components of vitamin E in group I also showed higher levels when compared to other groups. Group III was comprised of seven oil palm accessions, 1 from the H group, 1 from the M group, 3 from the L group and 2 from the EL group. The total vitamin E content in group III ranged from 171.30 to 1 258.50 ppm with an average of 532.94 ppm per accession. Both of group IV and V included two oil palm accessions and have an average of 543.23 and 345.49 ppm of total vitamin E content, respectively. Group II only contained one oil palm accession and the total vitamin E content was 256.29 ppm.

The correlation relationship between all markers and vitamin E content were analyzed. A total of 134 SNP and 7 indel markers showed significant ($p < 0.05$) correlation relationship with total vitamin E content. The associated SNPs were located onto 16 candidate genes involved in the vitamin E biosynthesis, including *EgCMK-1* (2 SNP markers), *EgHGGT* (3), *EgHPPD* (1), *EgGGPS-2* (1), *EgTAT-3* (2), *EgTAT-4* (3), *EgTAT-5* (2), *EgTAT-7* (3), *EgTAT-8* (6), *EgTAT-9* (16), *EgTAT-14* (6), *EgTAT-16* (4), *EgMPBQ-MT* (10), *EgTC* (5), *EgTMT-1* (67) and *EgTMT-3* (3) (S5 Table). The significantly associated indels only located onto *EgTAT16* (1) and *EgTMT-1* (6) (S6 Table). A large number of SNP (67) and indel (6) markers in *EgTMT-1* gene were significantly associated with the variations of vitamin E content.

PCR products amplified by *EgTMT-1-24* (*EgTMT-1*) primers contained two distinct alleles with amplicon sizes of 682 and 1 001 bp (Fig 5A). Sequencing results showed that the variation in the *EgTMT-1-24* locus was caused by a 319 bp indel in an intron region of *EgTMT* gene. While all 20 oil palm accessions contained the 682 bp allele, the other allele (1 001 bp) was detected in only eight oil palm accessions (R3, 4, 5, 7, 8, 9, 10, and 11), five of which (83%)

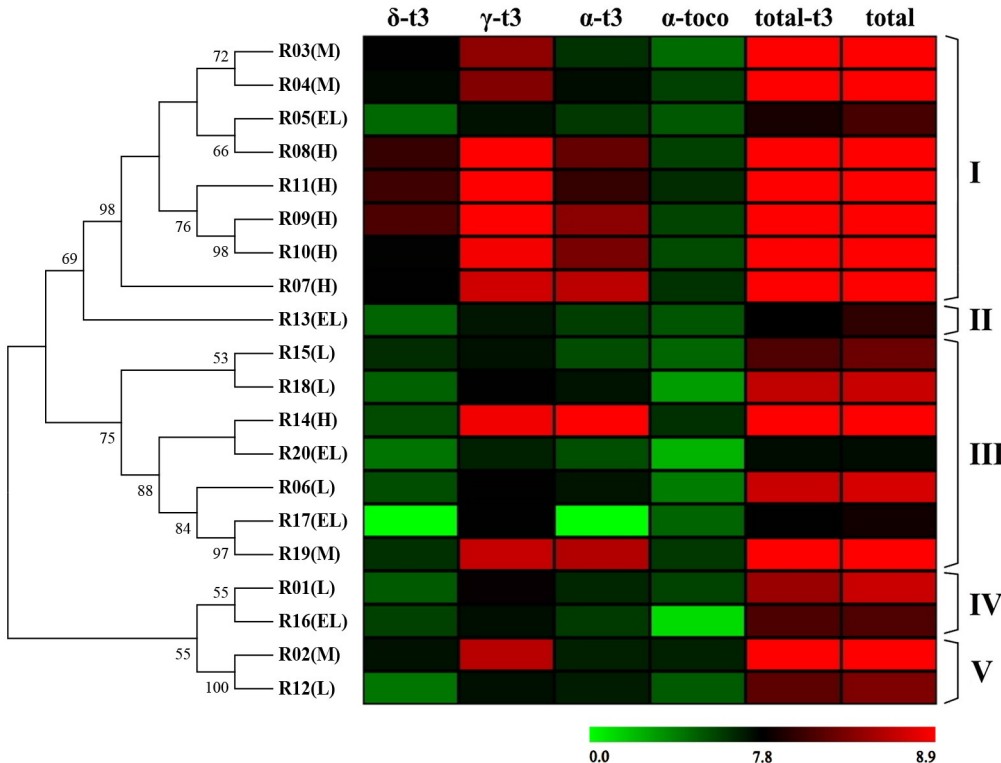

**Fig 4. The association between markers and vitamin E content.** The left diagram is the bootstrap consensus tree obtained by SNP and indel markers using the Maximum Likelihood method based on the General Time Reversible model with 500 bootstrap repetitions. Bootstrap values higher than 50% are shown. The heatmap was obtained by transformed vitamin E data (log₂ vitamin E data).

belonged to the H group (Fig 5B). In order to further validate the association between the *EgTMT-1-24* marker and vitamin E content, 53 oil palm accessions were amplified for this intron region with *EgTMT-1-24* primer pairs. Significant association relationships between *EgTMT-1-24* marker and total vitamin E content ($r = 0.504$), $\delta$-tocotrienol (0.661), $\gamma$-tocotrienol (0.511) and total tocotrienol content (0.506) were detected ($p$-value less than 0.001).

## The relationship between *EgTMT-1* expression and vitamin E content

Based on transcriptome data, the FPKM values of *EgTMT-1* gene were calculated in different oil palm individuals, including R17, R15, R12, R06, R04, R7, R9 and R14. The analysis showed

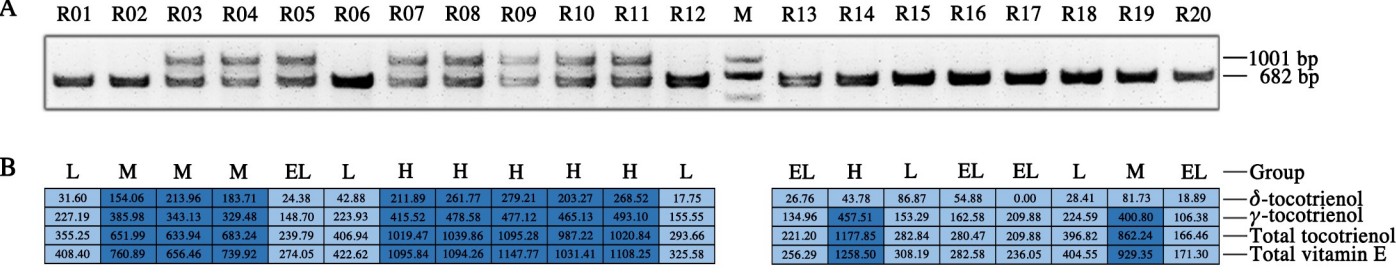

**Fig 5.** Electrophoresis diagram of PCR products amplified with the primer EgTMT-1-24 (A) and its association with vitamin E content (B).

that FPKM values in low vitamin E content individuals were 96.45 in R17, 80.10 in R15, 59.63 in R12, 61.37 in R06, and 63.01 in R04. While FPKM values in high vitamin E content individuals were 129.42 in R07, 106.01 in R09 and 131.82 in R14. The FPKM values of *EgTMT-1* had significant positive correlation with the total vitamin E contents ($r = 0.742$), $\gamma$-tocotrienol (0.737) and total tocotrienol content (0.737) at *p*-value less than 0.05. However, the correlation between FPKM values and $\delta$-tocotrienol was not significant ($r = 0.291$ and $p = 0.485$).

## Discussion

Palm oil is derived from the fresh oil palm fruit and widely used in food and oleochemical industry [84, 85]. Vitamin E is an essential component in palm oil which has powerful antioxidant activity and is useful for improving oil nutritional value [10, 86]. Consequently, it will be important to understand the vitamin E biosynthesis pathway, find key genes and identify functional markers for further developing improved oil palm varieties. By far, genes involved in vitamin E biosynthesis have been characterized and functionally validated in some higher plants, such as *A. thaliana* [8, 51, 87–90], *Oryza sativa* [91–93], *Glycine max* [94, 95], and *Z. mays* [96, 97]. However, the molecular basis of vitamin E biosynthesis in *E. guineensis* is was relatively unknown. In this study, we identified more candidate genes for vitamin E biosynthesis in *E. guineensis* (37) than in *A. thaliana* (18), *O. sativa* (23), and *Z. mays* (25). However, similar numbers of genes related to vitamin E biosynthesis were identified between *E. guineensis* (37) and *G. max* (39), which is interesting as both species have high vitamin E content. In *E. guineensis*, a significant expansion of the *EgTAT* gene family (17 copies) was detected relative to *A. thaliana* (3) [98], *O. sativa* (3), *Z. mays* (3), and *G. max* (5), which may suggest a higher efficiency of tyrosine transformation to *p*-hydroxyphenyl-pyruvate in oil palm.

Meanwhile, we identified one copy of *EgHGGT* and two copies of *EgHPT* that are crucial enzymes for tocotrienol and tocopherol biosynthesis, respectively. HGGT catalyzes HGA and GGDP into MGGBQ which is the precursor of tocotrienols, while HPT catalyzes HGA and PDP into MPBQ, which is the precursor of tocopherols [8]. Since tocotrienol comprises approximately 90% of the total vitamin E content in oil palm, *EgHGGT* should have a higher expression level or more gene copies in the genome of *E. guineensis*, which was not in accordance with our results. Therefore, it is possible that *EgHPT* genes can also catalyze HGA and GGDP to form MGGBQ in *E. guineensis*. *TMT* also showed higher copy number in *E. guineensis* (3) and *G. max* (3) [94, 95] compared to *A. thaliana* (1) [63, 99] and *O. sativa* (1) [92] and *Z. mays* (1) [97].

We also identified a large range in vitamin E content in the set of 20 oil palm accessions, with the total vitamin E content varying from 171.30 to 1 258.50 ppm, far larger than the variation observed in other species [100, 101]. Hence, we set out to ascertain gene sequence variation among these 20 oil palm accessions as an explanatory factor for this large variation in vitamin E content. Although most genetic variants (82% of SNPs and 89% of indels) were found in the intron regions of candidate genes, 226 SNPs were found in exons of candidate genes, and 56 of these SNPs changed amino acids in conserved protein domains. Research has shown that SNPs detected in the CDS and promoter sequences could cause noticeable phenotypical variation. In rice, one single nucleotide changes in reduced expression of C2H2-type transcription factor enhanced disease resistance [36]. The single nucleotide variation in *SHELL* gene, encoding a homologue of SEEDSTICK, influenced the oil yield in *E. guineensis* [102]. A single base substitution in *BADH*/*AMADH*, which causes amino acid change in a highly conserved motif, had lead to the variation of the existence of fragrance [103]. Therefore, these 56 SNPs may lead to the change of amino acid sequence in conserved domain and influence enzymatic activity and subsequent vitamin E biosynthesis.

## Conclusions

In summary, thirty-seven candidate genes involved in vitamin E biosynthesis were identified in oil palm. Gene structure, conserved domains and expression pattern of these candidate genes were analyzed. A total of 1 788 polymorphic sites (1 703 SNPs and 85 indels) were found among these 37 genes by Multiplex PCR sequencing and sequence alignment. Phylogenetic analysis based on all identified markers divided the 20 oil palm accessions into five groups which were related to the vitamin E content. Then the relationship between markers and vitamin E traits were further analyzed and a total of 141 function associated markers (134 SNP and 7 indel) were found. Among these markers, the 319 bp *EgTMT-1-24* indel located in an intron region of *EgTMT* gene was easily detected by PCR amplification and showed significant relationship with vitamin E content, especially tocotrienol content, which was validated in a total of 53 oil palm individuals. Expression analysis on the base of eight different oil palm individuals also revealed that the FPKM values of *EgTMT* gene had significant positive correlation with the vitamin E contents. These identified vitamin E functional markers are useful for further study of the key genes which could regulate vitamin E biosynthesis in oil palm mesocarp and may have potential application in marker-assisted selection breeding for high vitamin E content oil palm germplasm.

## Supporting information

**S1 Fig. SNP distributions across the CDS regions of the 37 candidate genes involved in vitamin E biosynthesis.** In the gene structure diagram, red blocks represent untranslated regions (UTR); pink blocks represent the conserved domain of the candidate genes; and the black lines represent SNP locations among the 37 candidate genes. Meanwhile, nucleotide and amino acid conversions were marked above the corresponding SNP markers.
(TIF)

**S1 Table. The detailed information of 20 oil accession used in the study.**
(XLSX)

**S2 Table. Primers sequences used in the study.**
(XLSX)

**S3 Table. Conserved domains of 37 candidate genes involved in vitamin E biosynthesis.**
(XLSX)

**S4 Table. Target gene length, amplified gene region, number of variations, total fragment length and coverage ratio of vitamin E biosynthesis genes.**
(XLSX)

**S5 Table. SNP markers across the 37 candidate genes involved in vitamin E biosynthesis and correlation coefficients with vitamin E content.**
(XLSX)

**S6 Table. The indels located in the 37 candidate genes involving vitamin E biosynthesis and correlation coefficients with vitamin E content.**
(XLSX)

**S1 Raw images. The original image of Fig 5A.**
(PDF)

## Author Contributions

**Conceptualization:** Wei Xia, Yong Xiao.

**Data curation:** Yajing Dou, Xiwei Sun.

**Formal analysis:** Yajing Dou, Wei Xia, Xiwei Sun, Haikuo Fan.

**Funding acquisition:** Yong Xiao.

**Project administration:** Wei Xia, Dongyi Huang, Yong Xiao.

**Supervision:** Annaliese S. Mason.

**Validation:** Annaliese S. Mason, Dongyi Huang, Haikuo Fan.

**Writing – original draft:** Yajing Dou, Yong Xiao.

**Writing – review & editing:** Yajing Dou.

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
