## [Decision Letter · Decision Letter 0]

23 Sep 2021

PONE-D-21-22518Developing functional markers for vitamin E biosynthesis in oil palmPLOS ONE

Dear Dr. Xiao,

Thank you for submitting your manuscript to PLOS ONE. After careful consideration, we feel that it has merit but does not fully meet PLOS ONE’s publication criteria as it currently stands. Therefore, we invite you to submit a revised version of the manuscript that addresses the points raised during the review process.

We look forward to receiving your revised manuscript.

Kind regards,

Maoteng Li

Academic Editor

PLOS ONE

2. We note that you are reporting an analysis of a microarray, next-generation sequencing, or deep sequencing data set. PLOS requires that authors comply with field-specific standards for preparation, recording, and deposition of data in repositories appropriate to their field. Please upload these data to a stable, public repository (such as ArrayExpress, Gene Expression Omnibus (GEO), DNA Data Bank of Japan (DDBJ), NCBI GenBank, NCBI Sequence Read Archive, or EMBL Nucleotide Sequence Database (ENA)). In your revised cover letter, please provide the relevant accession numbers that may be used to access these data. For a full list of recommended repositories, see http://journals.plos.org/plosone/s/data-availability#loc-omics or http://journals.plos.org/plosone/s/data-availability#loc-sequencing.

Additional Editor Comments (if provided):

Reviewers' comments:

Reviewer's Responses to Questions

**Comments to the Author**

1. Is the manuscript technically sound, and do the data support the conclusions?

Reviewer #1: Yes

Reviewer #2: Yes

2. Has the statistical analysis been performed appropriately and rigorously? 

Reviewer #1: Yes

Reviewer #2: Yes

3. Have the authors made all data underlying the findings in their manuscript fully available?

Reviewer #1: Yes

Reviewer #2: Yes

4. Is the manuscript presented in an intelligible fashion and written in standard English?

Reviewer #1: Yes

Reviewer #2: Yes

5. Review Comments to the Author

Reviewer #1: The authors discovered an impressive research, which showing twenty oil palm accessions with distinct variation of vitamin E contents (171.30 to1258.50 ppm), leading to the identification of 37 candidate genes involved in vitamin E biosynthesis in oil palm by using the known protein sequences of A. thaliana and Z. mays. Multiplex PCR sequencing for the 37 genes found 1703 SNPs and 85 indels among the 20 oil palm accessions, with 226 SNPs located in coding regions. This study identified a number of candidate function associated markers and provided clues for further research into molecular breeding for high vitamin E content oil palm. The results look quite promising. The statistical methods used in the study are good. The language and organization of paper are standardized. I would like to recommend to accept this manuscript. Thank you.

Reviewer #2: In the manuscript “Developing functional markers for vitamin E biosynthesis in oil palm”， the authors have detected 37 candidate genes associated with vitamin E synthesis in oil palm. The genes were sequenced and 134 SNPs and 7 indels were identified as functional markers that correlated with total vitamin E content. The molecular biology parts all sound. However, some revisions are still required before publication

1. The English need be edited

2. The raw data need be uploaded into the public database.

3. Line 47 change “Besides this” to “Besides”

4. Line 49 “which make palm oil be beneficial” should delete “…be..”

5. Line 63 “Such molecular markers, which are derived from polymorphic sites within genes

causally involved in phenotypic trait variation, are known as functional markers and are highly efficient in marker-assisted selection”, This sentence is confusing, could you change it?

6. Line 67 “and other approaches described elsewhere” you don’t need this.

Line 139 “similar sequences of potentially homologous genes in oil palm.” Delete “similar” here because you already mentioned the cut off value.

7. Line 257 “number of exons” change to “amount of exons”

8. Line 258 “same intron and exon number” change to “same amount of intron and exon”

9. Line 270 delete “each of”

10. Line 301 “each gene” change to “every gene”, there are some other lines with the same problem.

11. Line 321 ” Groups IV and V included two oil palm accessions each and averaged” change to “Both of group IV and V included two oil palm accessions and have an average of …”

6. PLOS authors have the option to publish the peer review history of their article (what does this mean?). If published, this will include your full peer review and any attached files.

Reviewer #1: **Yes: **Dan Qiu

Reviewer #2: **Yes: **LI Dongdong

---

## [Author Response · Author response to Decision Letter 0]

11 Oct 2021

Dear editors and reviewers,

Thank you very much for your comments and suggestions concerning our manuscript entitled “Developing functional markers for vitamin E biosynthesis in oil palm” (PONE-D-21-22518). Those comments are valuable and very helpful. We have made revisions according to reviewers’ comments carefully. Each comment raised by the editors and reviewers has been answered in the response sheet. I hope the revised version meets the requirements of “Plos One”. 

Details of how the comments have been addressed are prefixed by “>>>Response:” below the query to which they relate.

Reviewer #1:

The authors discovered an impressive research, which showing twenty oil palm accessions with distinct variation of vitamin E contents (171.30 to1258.50 ppm), leading to the identification of 37 candidate genes involved in vitamin E biosynthesis in oil palm by using the known protein sequences of A. thaliana and Z. mays. Multiplex PCR sequencing for the 37 genes found 1703 SNPs and 85 indels among the 20 oil palm accessions, with 226 SNPs located in coding regions. This study identified a number of candidate function associated markers and provided clues for further research into molecular breeding for high vitamin E content oil palm. The results look quite promising. The statistical methods used in the study are good. The language and organization of paper are standardized. I would like to recommend to accept this manuscript. Thank you.

>>>Response：Thank you for your comments. We really appreciate your efforts in reviewing our manuscript. Thank you again for agreeing to accept our manuscript.

Reviewer #2:

In the manuscript “Developing functional markers for vitamin E biosynthesis in oil palm”, the authors have detected 37 candidate genes associated with vitamin E synthesis in oil palm. The genes were sequenced and 134 SNPs and 7 indels were identified as functional markers that correlated with total vitamin E content. The molecular biology parts all sound. However, some revisions are still required before publication.

>>>Response：Thank you for your summary. We have revised the manuscript accordingly. Our point-by-point responses are listed below.

1. The English need be edited

>>>Response：Thank you for your suggestion. We have revised the manuscript carefully to improve the grammar and readability. The changes will not influence the content and framework of the manuscript and all the changes were marked in the revised manuscript. 

2. The raw data need be uploaded into the public database.

>>>Response：Thank you for your suggestion. The raw datas of multiple PCR sequencing of 20 oil palm individuals have been uploaded to European Nucleotide Archive (accession number is PRJEB38216, https://www.ebi.ac.uk/ena/browser/view/PRJEB38216). Meanwhile, the raw short reads datas of 24 transcriptomes (eight sample and three biological replicates) are aslo available in this database (accession number is PRJEB38102, https://www.ebi.ac.uk/ena/browser/view/PRJEB38102). 

3. Line 47 change “Besides this” to “Besides”

>>>Response：Thank you for your suggestion. We have changed “Besides this” to “Besides anti-oxidant” in the revised manuscript.

4. Line 49 “which make palm oil be beneficial” should delete “…be..”

>>>Response：Thank you for your suggestion. We have deleted the word “be” in this sentence in the revised manuscript.

5. Line 63 “Such molecular markers, which are derived from polymorphic sites within genes causally involved in phenotypic trait variation, are known as functional markers and are highly efficient in marker-assisted selection”, This sentence is confusing, could you change it?

>>>Response：Thank you for your suggestions. We have changed this sentence as “Polymorphic sites within genes that are related to trait variation could be used to develop functional markers, which are beneficial to marker-assisted selection” in the revised manuscript.

6. Line 67 “and other approaches described elsewhere” you don’t need this.

Line 139 “similar sequences of potentially homologous genes in oil palm.” Delete “similar” here because you already mentioned the cut off value.

>>>Response：Thank you for your suggestions. We have deleted the sentence “and other approaches described elsewhere” and the related references. 

We have changed the sentence “similar sequences of potentially homologous genes in oil palm” as “the best-hit homologous genes in oil palm” in the revised manuscript.

7. Line 257 “number of exons” change to “amount of exons”

>>>Response：Thank you for your suggestion. We have changed “number of exons” to “amount of exons” in the revised manuscript.

8. Line 258 “same intron and exon number” change to “same amount of intron and exon”

>>>Response：Thank you for your suggestion. We have changed “same intron and exon number” to “same amount of introns and exons” in the revised manuscript.

9. Line 270 delete “each of”

>>>Response：Thank you for your suggestion. We have changed the sentence “The functional domains were analyzed for each of the 37 candidate genes. All…” as “The functional domains analysis demonstrated that most candidate…” in the revised manuscript.

10. Line 301 “each gene” change to “every gene”, there are some other lines with the same problem.

>>>Response：Thank you for your suggestion. We have corrected “each gene” as “every gene” in the revised manuscript.

11. Line 321 “Groups IV and V included two oil palm accessions each and averaged” change to “Both of group IV and V included two oil palm accessions and have an average of …”

>>>Response：Thank you for your suggestion. We have changed “Groups IV and V included two oil palm accessions each and averaged” to “Both of group IV and V included two oil palm accessions and have an average of …” in the revised manuscript.

---

## [Editor Report · Decision Letter 1]

25 Oct 2021

Developing functional markers for vitamin E biosynthesis in oil palm

PONE-D-21-22518R1

Dear Dr. Xiao,

We’re pleased to inform you that your manuscript has been judged scientifically suitable for publication and will be formally accepted for publication once it meets all outstanding technical requirements.

Kind regards,

Maoteng Li

Academic Editor

PLOS ONE
---

## [Editor Report · Acceptance letter]

11 Nov 2021

PONE-D-21-22518R1 

Developing functional markers for vitamin E biosynthesis in oil palm 

Dear Dr. Xiao:

I'm pleased to inform you that your manuscript has been deemed suitable for publication in PLOS ONE. Congratulations! Your manuscript is now with our production department. 

Kind regards, 

on behalf of

Dr. Maoteng Li 

Academic Editor

PLOS ONE